# Improving inpatient provider-initiated HIV testing and counseling in Sierra Leone

**Getachew Kassa**[1]*, **Gillian Dougherty**[1], **Caitlin Madevu-Matson**[1], **Ginika Egesimba**[2], **Kenneh Sartie**[3], **Adewale Akinjeji**[2], **Francis Tamba**[3], **Brigette Gleason**[4], **Mame Toure**[2], **Miriam Rabkin**[1,5]

**1** ICAP at Columbia University, New York, NY, United States of America, **2** ICAP at Columbia University, Freetown, Sierra Leone, **3** Sierra Leone Ministry of Health and Sanitation, Freetown, Sierra Leone, **4** U.S. Centers for Disease Control and Prevention (CDC), Freetown, Sierra Leone, **5** Department of Medicine and Epidemiology, Columbia University Mailman School of Public Health, New York, NY, United States of America

* gk2353@cumc.columbia.edu

**Data Availability Statement:** All relevant data are within the paper and its Supporting Information files.

## Abstract

### Background/setting

Only 47% of HIV-positive Sierra Leoneans knew their status in 2017, making expanded HIV testing a priority. National guidelines endorse provider-initiated HIV testing and counselling (PITC) to increase testing coverage, but PITC is rarely provided in Sierra Leone. In response, a Quality Improvement Collaborative (QIC) was implemented to improve PITC coverage amongst adult inpatients.

### Methods

Ten hospitals received the intervention between October 2017 and August 2018; there were no control facilites. Each hospital aimed to improve PITC coverage to ≥ 95% of eligible patients. Staff received training on PITC and QIC methods and a package of PITC best practices and tools. They then worked to identify additional contextually-appropriate inter-ventions, conducted rapid tests of change, and tracked performance using shared indicators and time-series data. Supportive supervision bolstered QI skills, and quarterly meetings enabled diffusion of innovations while spurring friendly competition.

### Results

Baseline PITC coverage was 4%. The hospital teams tested diverse interventions using QI methods, including staff training; data review meetings; enhanced workflow processes and supervision; and patient education and sensitization activities Nine hospitals reached and sustained the 95% target, and all saw rapid and durable improvement, which was sustained for a median of six months. Of the 5,238 patients tested for HIV, 311 (6%) were found to be HIV-positive and were referred for treatment. HIV rapid test kit stockouts occurred during the project period, limiting PITC services in some cases.

**Funding:** ICAP at Columbia University received the award: This project was supported by the U.S. President's Emergency Plan for AIDS Relief (PEPFAR), through CDC cooperative agreement number U2GGH000994. Its contents are solely the responsibility of the authors and do not necessarily represent the official views of the Centers for Disease Control and Prevention or the Department of Health and Human Services. URL: https://www.cdc.gov/

**Competing interests:** The authors have declared that no competing interests exist.

## Conclusions

The intervention led to swift and sustained improvement in inpatient PITC coverage and to the diagnosis of hundreds of people living with HIV. Sierra Leone's Ministry of Health and Sanitation plans to take the initiative to national scale, with close attention to the issue of test kit stockouts.

## Introduction

The global scale-up of HIV services has enabled an estimated 21.7 million people to access antiretroviral treatment (ART) and contributed to a 42% reduction in deaths from AIDS-related illness and a 30% reduction in new HIV infections between 2010 and 2017 [1]. Population-based household surveys in multiple countries in sub-Saharan Africa suggest that the majority of people on ART have achieved viral suppression [2–4], but that not enough people know their HIV status [5]. This testing bottleneck is particularly evident in West and Central Africa, where Population-based HIV Impact Assessments (PHIA) show that only 46.9% and 37.2% of people living with HIV were aware of their status in Cameroon and Cote d'Ivoire respectively [6]. In order to achieve HIV epidemic control, countries will need to markedly improve testing coverage and subsequent linkage to treatment.

Sierra Leone's Ministry of Health and Sanitation (MoHS) has endorsed the Joint United Nations Program on HIV/AIDS (UNAIDS) targets for the year 2020, which call for 90% of all people living with HIV to know their status, 90% of those who know their status to initiate ART and 90% of those on ART to achieve durable viral suppression [7]. Unfortunately, access to HIV testing remains a significant challenge: in 2017, only 47% of people living with HIV in Sierra Leone knew their HIV status [8]. As in other countries, scaling up efficient and effective HIV testing and counseling services is critical.

Provider-initiated testing and counseling (PITC) is an evidence-based strategy to increase HIV testing coverage, in which health care workers (HCW) recommend HIV testing as a standard component of care for people attending health facilities. PITC targets a group of people who often have a higher HIV prevalence than the general population, have already come to the health system for help, and are relatively easy to reach, making it a core element of national HIV testing strategies. PITC has been endorsed by the World Health Organization (WHO) since 2007 [9], and subsequent studies confirm that it is an effective way to increase HIV testing coverage and yield in outpatient clinics, including general outpatient, antenatal and TB departments [10–14].

There are fewer studies of PITC for hospitalized patients, although this has been established as a productive strategy for pediatric inpatients in high HIV prevalence settings [15, 16]. In Sierra Leone, where adult HIV prevalence is estimated to be 1.5% [8], national guidelines support routine opt-out PITC for all inpatients, but the activity has not been prioritized. No national training resources, standard operating protocols or job aides are available for inpatient PITC, and inpatient HIV testing rates are not routinely monitored or evaluated.

Low PITC coverage rates are not unique to Sierra Leone [17]. Frequently-cited barriers to PITC in low-resource settings include lack of HCW training and skills, concerns about lack of privacy in crowded health facilities, provider workload, and HIV test kit stockouts [18–20]. Successful interventions to improve PITC coverage include the use of quality improvement (QI) methods, which bridge the "know-do" gap between established standards of care (what

we know) and the ability of health systems to implement evidence-based interventions and improve outcomes [21, 22].

In response to the missed opportunities to identify people living with HIV in Sierra Leone due to low PITC coverage, ICAP at Columbia University (ICAP) and the U.S. Centers for Disease Control and Prevention (CDC) partnered with MoHS to design and implement a QI Collaborative (QIC) to improve coverage of adult inpatient PITC at 10 hospitals in four districts. The purpose of our project was to build QI capacity and improve adult PITC coverage by designing and supporting a QI Collaborative (QIC) to catalyze swift improvement in PITC coverage for adult inpatients.

## Methods

### Project planning and design

The QIC approach is a well-defined improvement methodology, in which health facilities partner to address a quality challenge over a specified period of time [23]. QICs are quasi-experimental interventions that use time-series data to chart improvement over time, and typically do not use controls. A quality challenge is identified, along with a problem statement, an aim statement and shared indicators. QI teams at each health facility are trained and supported to conduct root cause analyses, identify contextually appropriate interventions and conduct rapid iterative tests of change using the Model for Improvement and its plan-do-study-act (PDSA) cycle (Fig 1) [24, 25]. Facility teams then come together for quarterly meetings, in which they compare progress and share interventions and innovations. Between learning sessions, QI teams receive twice-monthly site support and QI coaching visits. In addition to building QI capacity and improving outcomes, QICs also generally result in a "change package" of tools and approaches that can then be disseminated to additional health facilities [26].

In Sierra Leone, QIC preparations began in April 2017, and included stakeholder engagement, site selection, and development of aim statements and indicators. Given the limited experience with PITC in Sierra Leone, a "best practices toolkit" was also developed to share with health facility teams, and adjustments were made to ART recording and reporting tools to capture key information. ICAP also collaborated with MOHS to design project indicators (Table 1) and develop a monthly aggregate data collection form, standard operating protocols for project data collection, and a checklist to determine PITC eligibility, which was defined as having unknown HIV status (*e.g.*, not known to be HIV positive and not testing HIV negative within the past six weeks) and being able to give consent.

### Study setting

The health care facility in Sierra Leone is comprised of public, faith-based and private health sectors organized into three tiers of care: peripheral health units, district hospitals, and tertiary hospitals. Ten hospitals in four of Sierra Leone's 14 districts (Western Area Urban, Western Area Rural, Port Loko and Bombali) were purposively selected to participate based on their relatively high patient volume and to provide a diverse mix of public and private hospitals. Six were public hospitals (Blue Shield Hospital, King Herman Rd. Hospital, Lumley Hospital, Makeni Hospital, Police Hospital and Rokupa Hospital), three were run by faith-based organizations (Holy Spirit Hospital, St John of God Hospital and United Methodist Hospital) and one was private (Choithram Hospital). They had a median of 52 beds (range 20–180) and a median of 160 inpatient admissions per month (range 20–300).

In October and November 2017, five hospitals piloted the data collection tools and collected retrospective baseline data. Facility-level QI teams documented the number of people admitted, the number receiving PITC services, and the number testing positive in the four months

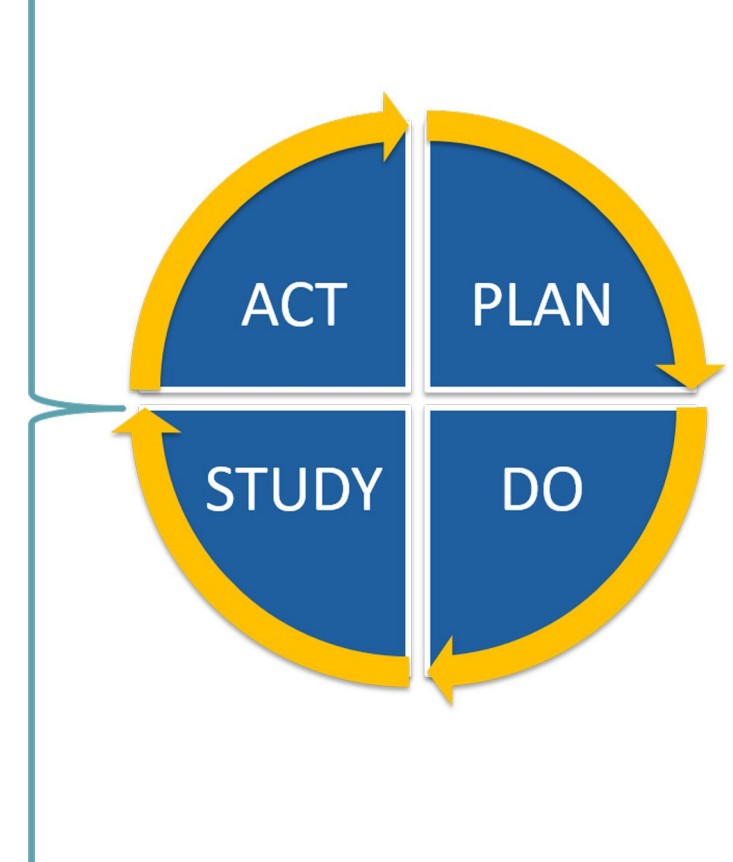

**Fig 1. Model for improvement: Adapted from: Langley G, Nolan T, Norman C, Provost L (1996).** The Improvement Guide. A practical approach to enhancing organizational performance.

from August—October 2017. These anonymized aggregate data showed that 35 of 850 adults admitted to participating hospitals during this time received PITC services, a baseline coverage of 4%. In consultation with MoHS and other stakeholders, a formal aim statement was developed for the QIC, with all hospitals working to achieve 95% PITC coverage in participating wards by the end of the 11-month project period. Each hospital implemented the project on four wards: male and female medical and surgical wards.

## Project implementation

From October to November 2017, ICAP and MoHS introduced the best practices toolkit and new project tools to each of the participating hospitals. QI teams were designated in each hospital, and included a focal person as well as healthcare workers from the HIV clinic and inpatient wards. Team structure varied from facility to facility, based on needs and context. In December 2017, QI teams from all ten hospitals came together to attend an initial five-day learning session at an off-site training venue. This workshop included training on PITC best practices and QI methods and tools, including the Model for Improvement and its PDSA cycle. A written pre- and post-test was used to assess participant knowledge. Participants

**Table 1. QI Indicators.**

| S. No. | Indicator | Numerator |
|---|---|---|
| | | **Denominator** |
| 1 | Proportion of eligible adult inpatients who received PITC during reporting month | # of adults receiving testing, counseling and results |
| | | # of adult inpatients eligible for PITC during reporting month |
| 2 | Proportion of adult inpatients assessed for PITC eligibility during reporting month | # of adult inpatients assessed for PITC eligibility |
| | | # of adults admitted to target wards during reporting month |
| 2 | Proportion of adult inpatients eligible for PITC during reporting month | # of adult inpatients eligible for PITC |
| | | # of adult inpatients assessed for PITC eligibility during reporting month |
| 3 | Proportion of inpatients testing positive for HIV during reporting month | # of adult inpatients testing positive during the reporting month |
| | | # adult inpatients receiving testing, counseling and results during reporting month |
| 4 | Proportion of newly-diagnosed HIV+ patients initiated on ART during admission | # newly diagnosed HIV+ inpatients discharged this month who initiated on ART during admission |
| | | # adult inpatients newly diagnosed with HIV during this admission discharged this month |
| 5 | # facilities experiencing HIV test kit stock-outs in reporting month | The total number of facilities that experienced HIV test kit stock outs |

completed 15 multiple-choice knowledge questions on the first day of the training (pre-test) before any course materials were introduced and on the final day of the training (post-test). Facility teams conducted root cause analyses using process maps and fishbone diagrams to determine why PITC was suboptimal on their inpatient wards. Each team worked to develop appropriate change ideas linked to their root cause analysis and returned to their hospital to test their initial change ideas using PDSA methods. The PDSA cycles facilitated iterative development and implementation of interventions. The PDSA approach enabled to test, rapid assessment and adapt the changes to ensure contextual interventions. Following the initial learning session, each hospital team worked to improve PITC coverage on four adult inpatient wards by testing change ideas and tracking monthly performance using the project indicators. ICAP and MoHS provided twice-monthly supportive supervision at each hospital, reviewing progress, deciding which change ideas were or were not working, and prioritizing new interventions when needed. Three additional learning sessions were convened in March, June and August of 2018, at which teams the 10 hospitals convened at an off-site training venue to review performance towards targets, describe their change ideas and lessons learned, and share tools and job aides. These quarterly meetings enabled participants to publicly compare their progress with their peer hospitals, fostering friendly competition and diffusion of innovations.

## Data collection, management and analysis

During the intervention period (October 2017 –August 2018), site-level QI teams collected aggregate performance data each month using standardized paper-based tools. ICAP staff entered the data into a project-specific online DHIS2 database instance that was systematically reviewed on a monthly basis for data quality. If errors or missing data were identified, hospital staff were contacted to obtain correct information. ICAP used DHIS2 to generate monthly descriptive statistics using a dashboard and graphs showing progress towards targets for each hospital, as well as the performance of the collaborative as a whole.

Aggregate data were analyzed monthly and at the conclusion of the project. QIC indicator performance was assessed for each hospital and for all 10 hospitals using trend analysis. The range, mean, and median across hospitals were also calculated for each indicator. Paired t-tests were used to compare the results of pre- and post-test scores at 0.05 significance level.

### Ethical review

The project received non-research determination approval from Columbia University's IRB (protocol #AAAR7670), the Sierra Leone Ethics and Scientific Review Committee, and the U. S. Centers for Disease Control and Prevention Center for Global Health office of the Associate Director for Science (tracking # 2019–051).

## Results

All ten hospitals participated in the QI project throughout the 11-month intervention period. The four QI learning sessions were well-attended, with an average of 50 participants attending each of the quarterly meetings. The initial training increased participant knowledge, as shown by improved performance on a multiple-choice test, from a median of 41% (29–47) at the pre-test to 73% (60–85) at the post-test (p < 0.0001). ICAP and MoHS provided twice-monthly supportive supervision, making a total of 188 site visits to the ten sites between October 2017 and August 2018.

QI teams used the Model for Improvement and PDSA methodology to identify and test contextually-appropriate change ideas to compliment the initial best practices toolkit, which included orientation and training slide sets, data registers and PITC eligibility checklists. Each team considered their root cause analysis, identified and prioritized interventions, and conducted rapid tests of change to see if the interventions improved performance. Barriers to PITC were found to be lack of prioritization by healthcare workers, scant documentation regarding PITC eligibility, inconsistency in the processes of PITC services, the sense that PITC was "no one's job" and negligible patient demand. Interventions included: staff training; monthly review meetings to assess data completion and quality; enhanced workflow processes and supervision; and patient education and sensitization activities (Table 2).

Nine of the ten hospitals reached and sustained the 95% target, and all hospitals saw rapid and durable improvement, which was sustained for a median of 6 (range 3–9) months out of the 11-month project period (Fig 2). As a result, 6,580 (79%) of the 8,329 patients admitted to participating wards during the project period were assessed for PITC eligibility, 5,668 (86%) were eligible, and 5,238 (92% of those known to be eligible and 63% of those admitted) received PITC services.

Of the 5,238 inpatients tested for HIV during the project period, 311 (6%) tested positive; of these, 197 (63%) started ART during their hospitalization and the rest were referred to the outpatient HIV department for ART initiation. All patients who tested negative received post-test prevention counseling. Teams also tracked supplies of HIV rapid test kits (RTK), anticipating that increased PITC volume might strain supplies. Six of the ten hospitals did report RTK stockouts by August 2018. Stockouts were reported in 15 of the cumulative 110 project-months of implementation, but all the facilities managed to replenish RTK supplies promptly when HCWs reported stockouts.

## Discussion

The ten hospitals achieved marked and sustained improvement in inpatient PITC coverage despite challenges with test kit stockouts. Of the patients admitted to participating wards with unknown HIV status who were determined to be eligible for PITC, 92% were tested and 6%

**Table 2. Change package–illustrative change ideas.**

| Change Idea | How the Change was Implemented |
|---|---|
| **Change Concept: Capacity building** | |
| Train hospital staff and senior management on PITC services, workflow and tools. | • Conduct in-service training for relevant staff and senior management with a focus on the importance of PITC, optimizing workflows, project registers and tools, and data collection SOPs |
| Orientation of newly-posted staff to PITC services | • Provide all newly-posted staff with orientation and training on PITC services focusing on pre- and post-test counseling techniques, patient flow processes, data flow, and data collection tools |
| On-the-job mentorship of ward nurses and other staff | • QI team lead provides mentorship to improve HIV testing and counseling, data flow and documentation |
| **Change Concept: Management & Leadership** | |
| Regular update on QI project to leadership | • QI team members hold monthly meetings on project implementation, update senior management staff on progress, and provide updates after every learning session |
| | • Identification and mitigation of missed opportunities to provide PITC services on inpatient wards (e.g., weekends, shifts, early-hour discharges) |
| Increasing PITC service time to include extended hours and weekends | • Sensitize and motivate facility staff to volunteer for PITC services by sharing data on PITC coverage to highlight the importance of full-week coverage |
| | • Train pool of volunteers to provide PITC services during evening shifts and weekends For example, each ward identified two PITC nurses during weekend shifts |
| Develop monthly schedule and duty roster to optimize PITC service in the wards | • Nursing leadership develops monthly staffing schedule and assigns PITC focal person responsibility |
| | • The ART clinic in-charge develops weekly duty rosters and provides mentorship to HIV counselors and ward nurses |
| Integration of HIV services into routine nursing services | • QI teams advocates with facility leadership to change PITC from a specialized service provided only by HIV clinic staff to a routine service provided by all nurses on all wards as part of routine care |
| | • QI team and HIV unit counselors provide orientation and mentorship to ward nurses building skills and confidence in delivering PITC |
| **Change Concept: Work Flow Process Review** | |
| Improving confidentiality of PITC services | • Provision of screens and space for private counseling |
| | • Encouragement of patients' relatives to wait outside while PITC services are provided |
| | • Patients occasionally moved to a private ward for PITC if no privacy available on general ward |
| Schedule inpatient testing prior to morning rounds every day | • HIV counselor visits the ward before the doctors' ward rounds to provide PITC services to eligible patients before they discharged home |
| Displaying a flow chart in every ward as a job aide to guide staff to use project tools | • Flow chart is developed and displayed to guide the delivery and documentation of PITC services |
| **Change Concept: Patient Engagement** | |
| Patient counseling and education on PITC | • HIV counselors and ward nurses develop lesson plans and health education messages on the importance of knowing one's HIV status, HIV transmission and prevention |
| | • Admission nurses provide HIV-specific health messages and handouts to patients admitted in the ward at the start of every shift |
| **Change Concept: Data Quality Assurance** | |

*(Continued)*

**Table 2.** (Continued)

| Change Idea | How the Change was Implemented |
|---|---|
| Monitoring and Evaluation | • QI team lead develops a weekly monitoring plan for wards, conducts random checks on ward PITC coverage, provides mentorship, and collates monthly data |
| Conduct monthly comprehensive data review | • The QI team meets monthly to review facility data quality and data management |
| | • Following data review, the team provides mentoring to HIV counselors and ward nurses to address identified gaps using QI methods and tools |

were found to be HIV positive. Hospital staff adapted existing best practices, implemented new job aides, screening tools and registers, and introduced QI methods to bridge the "know-do gap" and improve the processes and systems required to provide PITC services consistently and correctly. Of note, substantial improvement was achieved prior to the launch of QI testing and learning cycles, following the introduction of the initial PITC training and documentation tools. QIC methods then enabled sites to adapt the best practices toolkit to their specific contexts, and to sustain and institutionalize their early progress.

These findings are consistent with studies showing that the use of QI methods and the QIC approach can empower health facility teams to identify and address process and management barriers to attaining high quality health services [19, 20, 23, 27]. Contexts in which national guidelines and standards exist and quality challenges can be addressed at the facility level are optimal settings for QI interventions, which have been used successfully to address other quality challenges in Sierra Leone [28, 29, 30]. To our knowledge, this is the only study of QIC methodology to improve inpatient PITC coverage, but other PITC implementation projects report similar challenges, such as non-standardized workflows, suboptimal staffing and test kit shortages [10, 11, 31].

As is typical of QI projects, there were no control or comparison sites, and it is not possible to say which of the individual changes led to improvement. Study generalizability is also

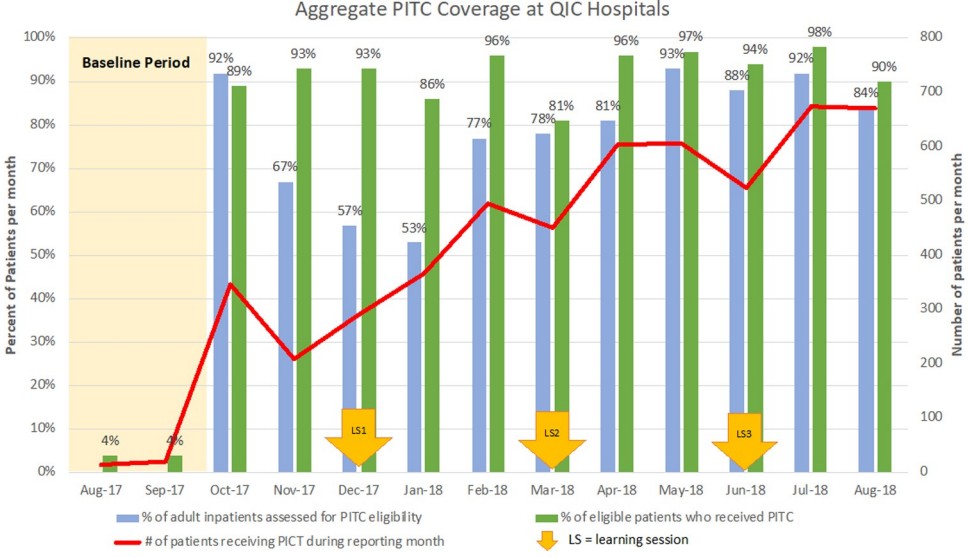

**Fig 2. Aggregate PITC coverage.**

limited by the purposive selection of participating health facilities and the focus on adult inpatients. Nonetheless, the rapid, large and sustained increase in PITC coverage strongly suggests that the tools and approaches used by participating hospitals can be transferred to other health facilities with good results. Given the success of the project and the relatively high testing yield, MoHS plans to leverage the tools and lessons learned to scale up inpatient PITC services nationwide.

## Supporting information

**S1 Appendix. Pretest/post-test results.**
(XLSX)

**S2 Appendix. Sierra Leone PITC data.**
(XLS)

## Acknowledgments

The authors acknowledge the support of the Sierra Leone Ministry of Health and Sanitation and the participating health facilities, and thank the Network of HIV Positives in Sierra Leone (NETHIPS) for their inputs.

## Author Contributions

**Conceptualization:** Getachew Kassa, Adewale Akinjeji, Brigette Gleason, Miriam Rabkin.

**Data curation:** Getachew Kassa, Ginika Egesimba.

**Formal analysis:** Getachew Kassa, Caitlin Madevu-Matson, Ginika Egesimba, Miriam Rabkin.

**Funding acquisition:** Mame Toure, Miriam Rabkin.

**Investigation:** Getachew Kassa, Gillian Dougherty, Adewale Akinjeji, Francis Tamba, Miriam Rabkin.

**Methodology:** Getachew Kassa, Ginika Egesimba, Miriam Rabkin.

**Project administration:** Getachew Kassa.

**Supervision:** Ginika Egesimba, Kenneh Sartie, Adewale Akinjeji, Francis Tamba, Brigette Gleason, Mame Toure, Miriam Rabkin.

**Validation:** Getachew Kassa, Caitlin Madevu-Matson, Miriam Rabkin.

**Visualization:** Getachew Kassa, Gillian Dougherty, Caitlin Madevu-Matson, Miriam Rabkin.

**Writing – original draft:** Getachew Kassa, Miriam Rabkin.

**Writing – review & editing:** Getachew Kassa, Gillian Dougherty, Caitlin Madevu-Matson, Ginika Egesimba, Kenneh Sartie, Brigette Gleason, Mame Toure, Miriam Rabkin.

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
