## [Decision Letter · Decision Letter 0]

6 May 2020

PONE-D-20-04669

Improving Inpatient Provider-Initiated HIV Testing and Counseling in Sierra Leone

PLOS ONE

Dear Dr Kassa,

Thank you for submitting your manuscript to PLOS ONE. After careful consideration, we feel that it has merit but does not fully meet PLOS ONE’s publication criteria as it currently stands. Therefore, we invite you to submit a revised version of the manuscript that addresses the points raised during the review process.

We would appreciate receiving your revised manuscript by Jun 20 2020 11:59PM. To enhance the reproducibility of your results, we recommend that if applicable you deposit your laboratory protocols in protocols.io, where a protocol can be assigned its own identifier (DOI) such that it can be cited independently in the future. For instructions see: http://journals.plos.org/plosone/s/submission-guidelines#loc-laboratory-protocols

We look forward to receiving your revised manuscript.

Kind regards,

Olanrewaju Oladimeji, MB;BS, Ph.D.

Academic Editor

PLOS ONE

Journal Requirements:

2. Please address the following:

a) Please include additional information regarding the survey or questionnaire used in the study and ensure that you have provided sufficient details that others could replicate the analyses. For instance, if you developed a questionnaire as part of this study and it is not under a copyright more restrictive than CC-BY, please include a copy, in both the original language and English, as Supporting Information.

b) Please ensure you have thoroughly discussed any potential limitations of this study within the Discussion section.

Additional Editor Comments (if provided):

There is a need to be very clear on the design of the study

Figue of a PDSA cycle will be helpful to understand and better appreciate the QI within the short period

Reference 1 is missing in the text

Please include the IRB approval numbers

Need to cite relevant supporting findings

Authors should use the COREQ checklist or other relevant checklists listed by the Equator Network such as the SRQR to ensure complete reporting

Reviewers' comments:

Reviewer's Responses to Questions

**Comments to the Author**

1. Is the manuscript technically sound, and do the data support the conclusions?

Reviewer #1: Yes

Reviewer #2: Partly

Reviewer #3: Yes

2. Has the statistical analysis been performed appropriately and rigorously? 

Reviewer #1: Yes

Reviewer #2: No

Reviewer #3: Yes

3. Have the authors made all data underlying the findings in their manuscript fully available?

Reviewer #1: Yes

Reviewer #2: Yes

Reviewer #3: Yes

4. Is the manuscript presented in an intelligible fashion and written in standard English?

Reviewer #1: Yes

Reviewer #2: Yes

Reviewer #3: Yes

5. Review Comments to the Author

Reviewer #1: Abstract

1. It will be better in the authors included a sentence on PITC as the opening statement of the abstract. To enhance the comprehensibility of the concept from the abstract as it is in the introduction.

2. It will be important for the authors to state clearly the objective(s) of this study. In as much as it was aimed at achieving quality improvement collaboration. The exact objective(s) of the study has not been clearly brought to light in the background/introduction.

3. It is also important for the authors to state clearly the study design used. It appears quasi experimental study/an interventional study with a before and after design. However, it has not been succinctly stated.

Introduction

1. Reference on line 49 needs to be revised

Methods

1. The authors have provided a detailed and elaborate narration on the QIC plan and design. However, since this was a study conducted in 10 hospitals, details of actual QIC plan and design were sketchy e.g. the composition of each facility team, the numbers of quarterly held, the number of monthly site support and QI coaching visits received in the methods section as against the results section etc.

Project implementation

1. It is also important to be clear about the designation of the head of each facility team as well as if the leaning sessions were conducted in facility or else where. Also, the authors need to provide information on measures put in place to ensure similarity and validity of the learning sessions contents if provided by different group of people.

Results

1. The authors should consider including the interquartile range of the median pre and post test scores of the participants for completeness sake. Furthermore, it is suggested that the authors carry out a statistical test of difference of median

scores (pre and post test) e.g wilcoxon signed rank test so as to have statistical evidence of improvement in the test scores following the training.

2. The authors should also consider providing information on demographic characteristic of the participants e.g. age, gender, highest education qualification, duration of practice and previous attendance of such training etc if the information were collected in the course of the study. This will go along way in enhancing the readers' understanding of the participants and as well as promoting reproducibility of QIC in other similar settings

3.Stock out of kits was reported by 6 of the 10 facilities but the authors did not mention any measures instituted to mitigate against stock out at the commencement of the study as well as the likely effects of the stock out on the final outcome for example in those facilities where stock out was experienced, what proportion of those inpatients who met the inclusion criterion was missed due to stock out etc.

Discussion

1.The authors may need to revise the discussion section making it more robust and exhaustive bringing to light the implications of the findings of the study in the light of other similar studies. Furthermore, it is important that the authors also bring to context how the findings of the study will impact practice in this setting and other and other settings alike.

References

1.It is important that the authors revise the references in line with the specified journal style e.g reference 6

Reviewer #2: Overall it is well written and straight forward. The following limitations are noted:

a) The research design could have been improved by using a stepped wedge cluster randomised controlled trial by having half of the hospitals start in the experimental condition while the rest would receive not intervention as control groups. The control group should have then receive a delayed intervention later on.

b) Not much is said about data analysis except for a presentation of a graph.

c) More importantly, the Discussion is rather skimpy.

- Previous studies elsewhere which have provided similar findings are not cited.

- There are no limitations to the findings presented.

Reviewer #3: The paper describes methods used in scaling up PITC in clear terms. The relevance of the study has been described. The methodologies adequately explained. There may be need to explain further on the one site that did not meet the 95% mark for introducing PITC as this may be a lesson especially for programs that will use the lessons in this for scaling up similar interventions

6. PLOS authors have the option to publish the peer review history of their article (what does this mean?). If published, this will include your full peer review and any attached files.

Reviewer #1: Yes: Afolaranmi Tolulope O

Reviewer #2: No

Reviewer #3: Yes: Dr Patrick Dakum

---

## [Author Response · Author response to Decision Letter 0]

4 Jun 2020

Editor Comments:

a) Please include additional information regarding the survey or questionnaire used in the study and ensure that you have provided sufficient details that others could replicate the analyses. For instance, if you developed a questionnaire as part of this study and it is not under a copyright more restrictive than CC-BY, please include a copy, in both the original language and English, as Supporting Information.

Response: A table with list of QI indicators used for collecting data attached.

b) Please ensure you have thoroughly discussed any potential limitations of this study within the Discussion section. Additional Editor Comments (if provided):

Response: Potential limitations are included.

C. There is a need to be very clear on the design of the study

Response: We added relevant detail to the methods section to emphasize that this is a non-experimental design that reports time-series data.

d. Figure of a PDSA cycle will be helpful to understand and better appreciate the QI within the short period

Response: Added to methods section

e. Reference 1 is missing in the text

Response: Added

f. Please include the IRB approval numbers

Response: The project received non-research determination from the Columbia University IRB, protocol AAAR7670. The 

g. CDC CGH HSR tracking # is 2019-051. Need to cite relevant supporting findings

Response: Added references especially in discussion section

Reviewer #1: 

Abstract

1. It will be better in the authors included a sentence on PITC as the opening statement of the abstract. To enhance the comprehensibility of the concept from the abstract as it is in the introduction. 

Response: We have edited the abstract for clarity.

2. It will be important for the authors to state clearly the objective(s) of this study. In as much as it was aimed at achieving quality improvement collaboration. The exact objective(s) of the study has not been clearly brought to light in the background/introduction.

Response: We have added this to the abstract. 

3. It is also important for the authors to state clearly the study design used. It appears quasi experimental study/an interventional study with a before and after design. However, it has not been succinctly stated.

Response: This is correct, the QIC used time series data to describe change over time; we added detail to the methods section to highlight this point. 

Introduction

1. Reference on line 49 needs to be revised

Response: Revised.

Methods

1. The authors have provided a detailed and elaborate narration on the QIC plan and design. However, since this was a study conducted in 10 hospitals, details of actual QIC plan and design were sketchy e.g. the composition of each facility team, the numbers of quarterly held, the number of monthly site support and QI coaching visits received in the methods section as against the results section etc.

Response: The team composition (HCWs from wards and counselors from HIV clinics), # of learning sessions and supportive supervisions is described and moved to method section.

Project implementation

1. It is also important to be clear about the designation of the head of each facility team as well as if the leaning sessions were conducted in facility or elsewhere. Also, the authors need to provide information on measures put in place to ensure similarity and validity of the learning sessions contents if provided by different group of people.

Response: Team structure varied from hospital to hospital.

Learning sessions were conducted out of the facility, bringing the 10 hospital teams together; all participants thus experienced identical training.

Results

1. The authors should consider including the interquartile range of the median pre and post test scores of the participants for completeness sake. Furthermore, it is suggested that the authors carry out a statistical test of difference of median scores (pre and post test) e.g wilcoxon signed rank test so as to have statistical evidence of improvement in the test scores following the training.

Response: Paired t test used and revised in the result section.

2. The authors should also consider providing information on demographic characteristic of the participants e.g. age, gender, highest education qualification, duration of practice and previous attendance of such training etc if the information were collected in the course of the study. This will go along way in enhancing the readers' understanding of the participants and as well as promoting reproducibility of QIC in other similar settings

Response: This data was not collected.

3.Stock out of kits was reported by 6 of the 10 facilities but the authors did not mention any measures instituted to mitigate against stock out at the commencement of the study as well as the likely effects of the stock out on the final outcome for example in those facilities where stock out was experienced, what proportion of those inpatients who met the inclusion criterion was missed due to stock out etc.

Response: Added few description on result and discussion section to clarify that testing kit stock outs experienced, but had no impact on the HIV testing coverage during the project period as HCWs engaged leadership and replenished the supply. But, highlighted the challenge as it would be a barrier for maintaining PITC services and HIV testing coverage.

Discussion

1.The authors may need to revise the discussion section making it more robust and exhaustive bringing to light the implications of the findings of the study in the light of other similar studies. Furthermore, it is important that the authors also bring to context how the findings of the study will impact practice in this setting and other and other settings alike.

Response: Revised the discussion considering other studies, and highlighted the significance of the findings.

References

1.It is important that the authors revise the references in line with the specified journal style e.g reference 6

Response: Revised.

Reviewer #2: 

Overall it is well written and straight forward. The following limitations are noted:

a) The research design could have been improved by using a stepped wedge cluster randomised controlled trial by having half of the hospitals start in the experimental condition while the rest would receive not intervention as control groups. The control group should have then receive a delayed intervention later on.

Response: This is correct. QI collaboratives are a proven approach to improving service delivery – they are not research studies and do not have an experimental design. 

b) Not much is said about data analysis except for a presentation of a graph.

Response: Data analysis methods are described on pp. 6-7 and results are described on pp. 8-9

c) More importantly, the Discussion is rather skimpy.

- Previous studies elsewhere which have provided similar findings are not cited.

- There are no limitations to the findings presented.

Response: We have enhanced the discussion and added limitations and references.

Reviewer #3: 

The paper describes methods used in scaling up PITC in clear terms. The relevance of the study has been described. The methodologies adequately explained. There may be need to explain further on the one site that did not meet the 95% mark for introducing PITC as this may be a lesson especially for programs that will use the lessons in this for scaling up similar interventions. 

Response: All the facilities improved but one of the facilities didn’t achieve 95% due to many factors including workload and change of trained staff.

---

## [Editor Report · Decision Letter 1]

10 Jun 2020

PONE-D-20-04669R1

Improving Inpatient Provider-Initiated HIV Testing and Counseling in Sierra Leone

PLOS ONE

Dear Getachew Belay Kassa, 

Thank you for submitting your manuscript to PLOS ONE. After careful consideration, we feel that it has merit but does not fully meet PLOS ONE’s publication criteria as it currently stands. Therefore, we invite you to submit a revised version of the manuscript that addresses the points raised during the review process.

Please remove degree after authors' names.

Include authors' contributions section.

use the PDSA to illustrate your implementation.

A brief description of the study setting in the methods section will be helpful.

Please submit your revised manuscript by June 24. If you will need more time than this to complete your revisions, please reply to this message or contact the journal office at plosone@plos.org. Please include the following items when submitting your revised manuscript:

We look forward to receiving your revised manuscript.

Kind regards,

Olanrewaju Oladimeji, MB;BS, Ph.D.

Academic Editor

PLOS ONE

Additional Editor Comments (if provided):

Please remove degree after authors' names.

Include authors' contributions section.

use the PDSA to illustrate your implementation.

A brief description of the study setting in the methods section will be helpful.

---

## [Author Response · Author response to Decision Letter 1]

24 Jun 2020

1. Please remove degree after authors' names. Response: Removed the degrees from the authors’ names in the manuscript

2. Include authors' contributions section. Response: Authors Contributions included at the end of the manuscript

3. Use the PDSA to illustrate your implementation. Response: Added a statement on PDSA under the subtitle ‘Project implementation’

4. A brief description of the study setting in the methods section will be helpful. Response: Added some statements to describe the study setting under Methods section.

Uploaded the response to reviewers, clean version of the manuscript and manuscript with track changes.

---

## [Editor Report · Decision Letter 2]

7 Jul 2020

Improving Inpatient Provider-Initiated HIV Testing and Counseling in Sierra Leone

PONE-D-20-04669R2

Dear Dr. Kassa,

We’re pleased to inform you that your manuscript has been judged scientifically suitable for publication and will be formally accepted for publication once it meets all outstanding technical requirements.

Kind regards,

Olanrewaju Oladimeji, Ph.D., MB; BS

Academic Editor

PLOS ONE

Additional Editor Comments (optional):

Accept
---

## [Editor Report · Acceptance letter]

13 Jul 2020

PONE-D-20-04669R2 

Improving Inpatient Provider-Initiated HIV Testing and Counseling in Sierra Leone 

Dear Dr. Kassa:

I'm pleased to inform you that your manuscript has been deemed suitable for publication in PLOS ONE. Congratulations! Your manuscript is now with our production department. 

Kind regards, 

on behalf of

Dr. Olanrewaju Oladimeji 

Academic Editor

PLOS ONE